# Marine Chitooligosaccharide Alters Intestinal Flora Structure and Regulates Hepatic Inflammatory Response to Influence Nonalcoholic Fatty Liver Disease

**DOI:** 10.3390/md20060383

**Published:** 2022-06-07

**Authors:** Jiayao Feng, Yongjian Liu, Jiajia Chen, Yan Bai, Jincan He, Hua Cao, Qishi Che, Jiao Guo, Zhengquan Su

**Affiliations:** 1Guangdong Engineering Research Center of Natural Products and New Drugs, Guangdong Provincial University Engineering Technology Research Center of Natural Products and Drugs, Guangdong Pharmaceutical University, Guangzhou 510006, China; fjy54525452@163.com (J.F.); yongjianliux@163.com (Y.L.); 15768996925@163.com (J.C.); 2Guangdong Metabolic Disease Research Center of Integrated Chinese and Western Medicine, Key Laboratory of Glucolipid Metabolic Disorder, Ministry of Education of China, Guangdong TCM Key Laboratory for Metabolic Diseases, Guangdong Pharmaceutical University, Guangzhou 510006, China; 3School of Public Health, Guangdong Pharmaceutical University, Guangzhou 510310, China; angell_bai@163.com (Y.B.); hejincan300@163.com (J.H.); 4School of Chemistry and Chemical Engineering, Guangdong Pharmaceutical University, Zhongshan 528458, China; caohua@gdpu.edu.cn; 5Guangzhou Rainhome Pharm & Tech Co., Ltd., Science City, Guangzhou 510663, China; cheqishi@rhkj.com.cn

**Keywords:** marine chitooligosaccharide, nonalcoholic fatty liver disease, intestinal flora, LPS/TLR4/NF-κB inflammatory pathway, short-chain fatty acid

## Abstract

In this study, C57BL/6 mice were given an HFHSD diet for 8 weeks to induce hepatic steatosis and then given COSM solution orally for 12 weeks. The study found that the HFHSD diet resulted in steatosis and insulin resistance in mice. The formation of NAFLD induced by HFHSD diet was related to the imbalance of intestinal flora. However, after COSM intervention, the abundance of beneficial bacteria increased significantly, while the abundance of harmful bacteria decreased significantly. The HFHSD diet also induced changes in intestinal bacterial metabolites, and the content of short-chain fatty acids in cecal contents after COSM intervention was significantly higher than that in the model group. In addition, COSM not only improved LPS levels and barrier dysfunction in the ileum and colon but upregulated protein levels of ZO-1, occludin, and claudin in the colon and downregulated the liver LPS/TLR4/NF-κB inflammatory pathway. We concluded that the treatment of marine chitooligosaccharide COSM could improve the intestinal microflora structure of the fatty liver and activate an inflammatory signaling pathway, thus alleviating the intrahepatic lipid accumulation induced by HFHSD.

## 1. Introduction

Nonalcoholic fatty liver disease (NAFLD) is the most common chronic liver disease [1]. A meta-analysis conducted by Younossi et al. [2] showed that the prevalence of NAFLD reached as high as 25% worldwide, and high prevalence of this disease has been reported in North America (24%), Europe (24%), and Asia (27%). These data indicate that NAFLD has become one of the major public health problems in the world [3]. The main treatment methods for NAFLD currently include the following: (1) lifestyle changes; (2) drug therapy with pioglitazone, silymarin, and liraglutide, among other drugs; and (3) surgical intervention [4]. However, all of the aforementioned treatment methods have limitations, and additional treatment methods, such as drugs with few side effects and good efficacy, urgently need to be developed for the treatment of NAFLD.

Marine resources have the potential to be used in the development of high-value products. Polysaccharide biomaterials, represented by chitin and chitosan, collagen-based materials derived from marine biological composites, have been studied extensively [5]. Chitooligosaccharide (COSM) is a cationic, basic oligosaccharide with a degree of polymerization between 2 and 20 that is formed by linking glucosamine with n-acetyl glucose through a β-1, 4-glycosidic bond [6]. COSM has antibacterial, antiobesity, hypolipidemic, anti-inflammatory, hypoglycemic, antioxidant, antihypertension, anti-Alzheimer’s disease, and antitumor physiological functions [7,8]. Previous studies by this group showed that COSM could improve weight gain, lipid levels in the fat pad, liver steatosis, organ indexes, and other indicators of obesity in obese SD rats and C57BL/6 mice [9,10]. Moreover, COSM could protect against obesity by improving the lipolysis, thermogenesis, and insulin resistance (IR) of adipose tissue in obese mice [11]. However, recent studies have shown that COSM may exert metabolic benefits by reshaping the intestinal microecology and improving the intestinal barrier function [11,12,13].

The gut microbiome is a type of Intestinal microecology dominated by bacteria [14]. The total number of genes in the gut microbiome is hundreds of times higher than the number of genes in humans; thus, these bacteria can encode degradation enzymes that are lacking in most hosts, such as bile acid-producing hydrolases and enzymes that degrade carbohydrates that cannot be digested by hosts [15]. The intestinal flora and its metabolites play important roles in host metabolism. Studies have shown that “hepatointestinal axis” disturbances, such as intestinal barrier dysfunction, bacterial translocation, inflammatory response, and Toll-like receptor (TLR) signaling pathway activation, play key roles in the pathogenesis of NAFLD [16,17,18].

Therefore, by analyzing the abundance, structure, and metabolites of intestinal flora in the liver, we finally demonstrated that COSM with average molecular weight ≤3000 could improve hepatic steatosis through intestinal flora regulation of the liver LPS/TLR4/NF-κB inflammatory pathway. This study provided important clues for the molecular mechanism of COSM on intestinal microflora abnormalities and inflammatory responses in NAFLD, discussed the potential application prospects of the marine shell-like oligosaccharide COSM in drugs, and provided certain reference for the development of marine drugs.

## 2. Results

### 2.1. Changes in Body Weight, Food Intake, and Serum Glucose and Lipid Levels

During 12 weeks of oral administration of COSM, there were no significant differences in the body weights of HFHSD-induced fatty liver mice (Figure 1A). In addition, after 12 weeks of oral administration of COSM, the weight gain of each group showed that COSM did not affect the weight change of HFHSD mice (Figure 1B) (*p* > 0.05). It has been documented that weight loss is not necessarily an indicator of NAFLD improvement [19,20]. In addition, during 12 weeks of oral administration of COSM, there was no significant difference in food intake between the HFHSD group and other COSM groups (*p* > 0.05), and the intake of the HFHSD group was lower than that of the control group. These results suggest that the HFHSD, with a higher energy source, reduced appetite in NAFLD mice, while COSM did not affect appetite in NAFLD mice (Figure 1C). Previous studies [21] have shown that a higher proportion of Firmicutes compared with Bacteroidetes in the intestinal flora, i.e., a higher F/B ratio, can promote increase in energy intake, which may lead to an increase in fat storage in the body. Increased fat storage can lead to liver involvement and metabolic disorders and can increase the development of fatty liver. Therefore, as described in Section 2.5.2, a high F/B ratio could explain why the mice in the control group ate more food but had lower body weight, whereas the mice in each treatment group exhibited lower food intake but higher body weight compared with the mice in the control group. This result occurred because a high F/B ratio leads to high energy intake and increases in the appetite of mice and thus affects the body weight [22]. After 12 weeks of oral administration of COSM, an oral glucose tolerance test (Figure 1D–F) was performed, and serum glucose levels (Figure 1G) were measured. It was found that COSM-H treatment significantly improved hyperglycemia in HFHSD-fed mice, whereas the COSM-M and COSM-L treatments exerted poor effects. All doses of COSM could improve the high insulin level induced by an HFHSD (Figure 1H), and thus, COSM could improve IR and hyperglycemia.

### 2.2. COSM Could Improve the Lipid Content in the Serum and Liver of HFHSD-Fed Mice

The serum TG (triglyceride) levels in the mice with NAFLD were significantly higher in the metformin and COSM-H groups, and the serum TC (total cholesterol) levels in the mice with NAFLD were significantly higher in the metformin, COSM-H, and COSM-M groups (Figure 2A,B). COSM-H improved the serum levels of low-density lipoprotein cholesterol (LDL-C) and high-density lipoprotein cholesterol (HDL-C) in mice with NAFLD (Figure 2C,D). Both COSM and metformin reduced the level of free fatty acids (FFAs) (Figure 2E). These results suggest that COSM can improve the abnormalities in lipid metabolism caused by an HFHSD to a certain extent. The liver TG and TC levels were determined, and the results are shown in Figure 2F,G. The HFHSD significantly increased the liver TG and TC levels in the model group, which suggested serious accumulation of blood lipids in the model group. The TG and TC levels in the metformin, COSM-H, and COSM-M groups were significantly reduced, which indicated that all three interventions reversed the accumulation of liver lipids in the mice with NAFLD.

The livers of the mice in each group were observed (Figure 2H). The livers of the mice belonging to the control group were dark red in color, with obvious outlines and no yellow fat spots on their surface. In contrast, the livers of the mice in the model group were tawny with complete outlines. Observation revealed oil stains on the incision surface and obvious fat spots on the surface. The analysis of the COSM-treated groups revealed that the livers of the COSM-H-treated mice were notably similar to those of the control mice, as demonstrated by a dark red color, normal volumes, and occasional fat spots. The livers of the mice in the metformin group were more similar to those of the control mice, as demonstrated by a dark red overall appearance and fewer fat particles on the surface. COSM can improve hepatic steatosis in HFHSD-fed mice to a certain extent.

In this study, H&E staining, oil red O staining, and other pathological sections were used to verify the effect of COSM on liver fat accumulation in mice. Liver staining revealed virtually no lipid accumulation in the control livers. These livers also exhibited large fat vacuoles and loose cytoplasm, which was suggestive of severe steatosis (Figure 2I). Compared with the results found for the model group, significant reductions in lipid accumulation were observed in the livers of the metformin group and COSM-treated groups (particularly the COSM-H group). The oil red O staining results showed (Figure 2J) redder and larger lipid droplets in the livers of the model group than in those of the control group, whereas the numbers of lipid droplets in the livers of the metformin group and the COSM groups (particularly the COSM-H group) were significantly lower than that in the livers of the model group. The aforementioned experimental results indicated that COSM could significantly improve the accumulation of lipids in the liver induced by HFHSD feeding.

### 2.3. COSM Could Improve the Liver Function and Antioxidant Capacity of HFHSD-Fed Mice

Liver injury ensues when mitochondria are damaged and cell necrosis occurs. When this happens, aspartate aminotransferase (AST) and alanine aminotransferase (ALT) are released into the blood circulation, leading to increased serum AST and ALT levels, which can be considered indexes of liver injury and exert effects on treatment and prognosis. The serum AST and ALT levels in model group were 48.70% and 72.67% higher than those in the control group, respectively, suggesting that long-term HFHSD feeding may have damaged the liver function of the model group to a certain extent. COSM treatment improved the AST level compared with that of the model group, but the effect was not significant. The metformin, COSM-H, and COSM-M treatments significantly reduced the serum ALT levels of the mice with NAFLD (Figure 3A,B). These results suggest that COSM can improve liver dysfunction induced by HFHSD feeding to a certain extent.

The results from the assessment of serum antioxidant function showed that the HFHSD significantly decreased two serum antioxidant function indexes, namely the catalase (CAT) level and the total antioxidant capacity (T-AOC), in mice with NAFLD and that COSM increased the CAT level in mice with NAFLD to a certain extent. However, the results were not significant, and all the doses of COSM increased the serum T-AOC level (Figure 3C,D). These results suggest that COSM has a certain antioxidant function in the comprehensive ability of CAT and T-AOC, thus improving the oxidation level of NAFLD mice.

### 2.4. COSM Could Improve the Serum Inflammatory Response of HFHSD Mice

The NAFLD process is often accompanied by inflammation. The evaluation results of serum inflammatory factors in this study are shown in Figure 4A–C, and the results of gene studies at the liver level are shown in Figure 4D–F. TNF-α and IL-6 levels in serum and expression levels of liver genes in the model group were significantly higher than those in the control group, indicating that long-term HFHSD could cause systemic inflammation in mice. Metformin, COSM-H, and COSM-M significantly reduced the levels of TNF-α in the serum and liver of NAFLD mice as assessed by analysis of positive drug-treated mice and other COSM treated mice. Metformin could significantly reduce the serum IL-6 level but had no significant effect on the liver IL-6 gene level. COSM-H could significantly reduce serum and liver IL-6 gene levels. COSM-M induced no significant difference in serum IL-6 level but could reduce the level of the IL-6 gene in the liver. These results suggest that COSM and metformin can reduce the inflammatory response in HFHSD mice to a certain extent. The IL-10 levels in the serum and IL-6 gene levels in the liver in the COSM-H and COSM-M groups were significantly higher than those in model mice, suggesting that COSM treatment can promote the release of anti-inflammatory factors and ultimately achieve the goal of reducing inflammation.

### 2.5. Effects of COSM Intervention on Microflora and the Intestinal Tract

#### 2.5.1. COSM Could Improve the Intestinal Microfloral Structure of HFHSD-Fed Mice

According to the above-described analyses of serum, liver, and other indicators, the effect of the COSM-H treatment was better than those of the COSM-M and COSM-L treatments. Therefore, fecal samples from the COSM-H group were selected for the following analysis of the intestinal flora. In this section, the “COSM group” refers to the “COSM-H group”. To explore the different species among the groups, species showing differences between the model group and the control group and the model group and the COSM group were identified by *t*-test. The model group had significantly higher levels of *Lactococcus lactis* (*p* = 0.004), *Faecalibaculum rodentium* (*p* = 0.028), *Desulfovibrio* sp. *ABHU2SB* (*p* = 0.025), *Lachnospiraceae bacterium DW59* (*p* = 0.015), and *Clostridium* sp. *culture jar-13* (*p* = 0.014) than the control group. The levels of *Erysideras subtilis* (*p* = 0.001), *Bacteroides uniformis* (*p* = 0.026), *Bacteroides acidifaciens* (*p* = 0.047), *Clostridium paraputrificum* (*p* = 0.020), and *Bacteroides caecimuris* (*p* = 0.005) in the COSM group were significantly higher than those in the model group. The levels of *Desulfovibrio* sp. *ABHU2SB* (*p* = 0.044), *Lachnospiraceae bacterium DW59* (*p* = 0.020), and *Clostridium* sp. *culture jar-13* were significantly reduced by the COSM treatment (*p* = 0.026) (Figure 5A).

In conclusion, COSM significantly increased the levels of beneficial bacteria, such as *Bacteroides uniformis*, *Bacteroides acidifaciens*, and *Clostridium paraputrificum*, and decreased the levels of *Desulfovibrio* sp. *ABHU2SB* (*p* = 0.044), *Lachnospiraceae bacterium DW59* (*p* = 0.020), and other harmful bacteria. Thus, COSM could improve the intestinal microflora structure and may thereby eventually play a role in the treatment of fatty liver.

#### 2.5.2. COSM Could Inhibit the Growth of Harmful Intestinal Bacteria and Improve the Abundance of Dominant Bacteria in HFHSD Diet-Fed Mice

Alpha diversity, as characterized by the ACE, Chao1, Shannon, and Simpson indexes, is a major indicator of community richness and diversity. In general, greater values of these indicators indicate greater richness or diversity of the community [23,24]. The results from the α diversity analysis, which was performed by clustering each group of mouse feces, showed that the ACE, Chao1, Shannon, and Simpson indexes of all the groups were higher than those of the control group (Figure 5B), which suggested that HFHSD feeding increased the diversity of the intestinal microbiota of mice. Compared with that in the model group, the intestinal microflora diversity in the COSM group was decreased to a certain extent. The COSM treatment partially reversed the increases in the intestinal flora diversity and richness caused by HFHSD feeding.

Beta diversity analysis is typically performed to compare the composition of the microflora between different groups and thus evaluate the existence of differences in the microflora between groups. PCoA, principal component analysis (PCA), and nonmetric multidimensional calibration (NMDS) were performed. The results revealed differences between individuals or groups (Figure 5C). In the bottom figure, each point represents a sample, and points of the same type represent samples belonging to the same group. According to the PCoA, PCA, and NMD analyses, the sample distance between the model and control groups was greater after HFHSD feeding, which indicated a significant difference in the colony composition between the two groups. However, the microflora composition of the COSM group was closer to that of the model group than to that of the control group, which suggested that the COSM intervention alleviated the harmful effects of the HFHSD on the intestinal microflora composition to a certain extent. However, the overall adjustment remained similar to that of the model group.

Subsequently, the DNA of fecal samples from each group was sequenced, and a total of 1,923,156 high-quality clean reads were obtained after the original data obtained by the IonS5TMXL platform were spliced and subjected to quality control and chimeric removal; of these reads, 480,561 effective sequences were obtained for the control group, with an average of 80,093.5. In addition, 480,758 valid sequences, with an average of 80,126.33 sequences; 480,841 valid sequences, with an average of 801,40.17 sequences; and 480,996 valid sequences, with an average of 801,66 sequences, were obtained for the model, metformin, and COSM groups, respectively. The obtained effective sequences were clustered into OTUs with 97% consistency, and a total of 670 OTUs, including 504 OTUs in the control group, 522 OTUs in the model group, 527 OTUs in the metformin group, and 512 OTUs in the COSM group, were obtained. The OTUs of each group are shown in Figure 5D. According to the results from the OTU annotation, the fecal microbiota of the mice in each group were analyzed at the phylum and genus levels. Based on the top ten most abundant bacteria and the relative abundance histogram, the proportion of each bacterium in each sample group was obtained.

The analysis at the phylum level (Figure 5E) showed that *Bacteroidetes*, *Firmicutes*, and *Proteobacteria* exhibited the highest abundance in the fecal flora of the mice in each group and were the dominant flora in the intestinal flora. Compared with the control, HFHSD feeding significantly increased the abundance of *Firmicutes* and *Proteobacteria* (*p* < 0.001, *p* < 0.01) and significantly decreased the abundance of *Bacteroides* (*p* < 0.0001). Compared with the model group, the COSM treatment group exhibited a significantly increased abundance of *Bacteroidetes* (*p* < 0.01) and no significantly decreased abundance of *Firmicutes* (*p* = 0.0816) (Figure 5G). Previous studies [21] have shown that a higher proportion of *Firmicutes* compared with *Bacteroidetes* in the intestinal flora, i.e., a higher F/B ratio, can promote increase in energy intake, which may lead to an increase in fat storage in the body. Increased fat storage can lead to liver involvement and metabolic disorders and can increase the development of fatty liver. Therefore, the F/B ratio of the fecal microflora in each group was analyzed in this study. The F/B ratio of the model group was significantly higher than that of the control group (*p* < 0.001), which suggested that the HFHSD could promote the development of an intestinal flora structure that favored metabolic disorders. The COSM group exhibited a significantly lower F/B ratio than the model group (*p* < 0.01) (Figure 5F). This result was also consistent with the weight and food intake results described in Section 2.1.

The results from the genus-level analysis are shown in Figure 5H. The dominant genera were quantitatively analyzed (Figure 5I). COSM significantly increased the abundance of *Erysipelatoclostridium*, *Bacteroidetes*, and *Akkermansia* and significantly decreased the abundance of *Lactococcus*. Studies have shown that *Akkermansia* can promote GLP-1 secretion into the gastrointestinal tract of HFD-fed mice by secreting the P9 protein, which regulates the host’s energy balance, improves glucose homeostasis, promotes brown fat thermogenesis, reduces body weight [21], and significantly reduces immune-mediated inflammation and hepatocyte death [25]. The results obtained herein suggest that COSM can increase the abundance of probiotics, block the growth of harmful bacteria, and exert metabolic benefits.

#### 2.5.3. COSM Could Increase the Levels of Acetic Acid, Propionic Acid, and Total SCFAs in the Cecal Contents of HFHSD-Fed Mice

SCFAs play an important role in human metabolism [26]. Acetic acid accounts for the highest proportion of short-chain fatty acids produced by intestinal bacteria. It can adjust the pH value of the intestinal tract to allow beneficial bacteria to survive, prevent harmful bacteria, and help maintain the stability of the intestinal environment [27]. Furthermore, studies have confirmed that SCFAs have important research significance for NAFLD [28]. As shown in Figure 6A, COSM significantly increased the total SCFA content (*p* < 0.05) and significantly increased the levels of acetic acid (*p* < 0.05) and propionic acid (*p* < 0.01). Moreover, COSM reversed the changes in the butyric acid, isobutyric acid, isovaleric acid, and valeric acid levels induced by HFHSD, but the differences were not significant.

In conclusion, COSM significantly improved acetate, propionate, and total SCFA levels in the cecal contents of HFHSD-fed mice.

#### 2.5.4. COSM Could Improve the Villus Structure of the Ileum and the Integrity of Colonic Goblet Cells and the Mucosal Layer of HFHSD-Fed Mice

To investigate the effect of COSM on the HFHSD-induced changes in the intestinal barrier structure, H&E-stained paraffin sections of the ileum and colon were pathologically analyzed. The results from the ileum, which is the last part of the small intestine, are shown in Figure 6B. The ileum crypt depth of the model group was significantly reduced compared with that of the control group, and damage was observed in the mice of the model group. The villus height and crypt depth of the ileum of the mice in the COSM group were significantly increased compared with those of the mice in the model group. These results indicate that the HFHSD could damage the structure of the villi in the ileum of model mice and that COSM treatment could effectively improve the structure of the villi in the ileum of model mice. Compared with those of the control mice, the colon goblet cells and part of the mucosal layer of the model mice were destroyed. COSM obviously protected the integrity of goblet cells and the mucus layer, and the COSM-H treatment exerted stronger effects (Figure 6C).

### 2.6. COSM Improved the Intestinal Wall Barrier Integrity and Endotoxemia in HFHSD-Fed Mice

LPS is a metabolite of intestinal bacteria, and increased levels of LPS in the serum indicate that the intestinal barrier is obstructed, which allows greater amounts of LPS to enter the blood circulation through this barrier [29,30]. Previous studies have shown that an HFHSD can increase intestinal permeability and reduce tight junction protein expression; subsequently, the bacterial metabolite LPS penetrates the intestinal barrier to enter the blood circulation, resulting in metabolic endotoxemia and metabolic disorders [31]. The serum LPS levels in the mice were measured, and the results showed that the serum LPS levels in the model group were significantly higher than those in control group, which indicated that HFHSD feeding significantly increased the translocation of LPS to the blood circulation. Metformin and COSM-H significantly reduced the serum LPS levels compared with those in the model group (*p* < 0.001, *p* < 0.05; Figure 7A).

Occludin, ZO-1, and claudin are important components for maintaining the intestinal epithelial barrier [32,33]. In this experiment, the expression of occludin, ZO-1, and claudin in colon tissues of mice was measured by WB. Our results showed that the protein expression levels of ZO-1, occludin and claudin in the model group were lower than those in the control group (*p* < 0.05, *p* = 0.1037, *p* = 0.0857), which indicated that the HFHSD could reduce the expression of barrier-related proteins in the mouse colon, resulting in intestinal barrier dysfunction. After the intervention, the ZO-1 level in the COSM group was significantly higher than that in the model group (*p* < 0.05); the COSM treatment also increased the occludin level, but the increase was not significant (*p* = 0.1084), and the level of claudin was not significantly changed by the treatment (Figure 7B–E). In addition, the immunofluorescence staining results showed that COSM administration reversed the decrease in colonic ZO-1 protein fluorescence (Figure 7F). These results suggest that metformin and COSM can improve the mechanical barrier function of the intestinal mucosa, improve the integrity of the intestinal wall barrier, and improve endotoxemia.

### 2.7. COSM Improved the LPS/TLR4/NF-κB Signaling Pathway in the Livers of HFHSD-Fed Mice

The HFHSD resulted in disruption of the intestinal mucosal barrier and intestinal leakage in mice, which led to increased serum LPS levels [34]. LPS activates the TLR4 receptor on the surface of liver cells and subsequently the downstream signaling molecule MyD88, resulting in the phosphorylation of IκB and its separation from NF-κB. NF-κB is then phosphorylated and migrates to the nucleus as part of the TLR4/MyD88/NF-κB signaling pathway, which triggers a series of inflammatory processes. This pathway can cause IR in hepatic Kupffer cells and promote fibrosis in hepatic stellate cells. A WB analysis showed that HFHSD feeding upregulated the expression of TLR4 and significantly increased the protein expression of P-NF-κB S536 (*p* < 0.001) but did not significantly change the level of NF-κB. COSM intervention significantly decreased the P-NF-κB S536 protein expression level (*p* < 0.05) and did not significantly decrease the protein expression level of TLR4, but there were certain trends (*p* = 0.1996) compared with the levels found in the model group (Figure 8A–D). RT-PCR analysis showed that HFHSD increased the expression of the TLR4 and NF-κB S536 genes. COSM intervention did not significantly reduce expression of the TLR4 or NF-κB S536 genes (*p* < 0.05), but overall, COSM showed a certain improvement trend compared with the model group (Figure 8A–D). COSM could improve colonic barrier function, reduce intestinal leakage, and reduce the serum LPS levels (Figure 7A) and thereby reduce the expression levels of components of the liver LPS/TLR4/NF-κB signaling pathway; thus, COSM may ultimately play a therapeutic role in improving liver disease in mice with NAFLD.

## 3. Discussion

Marine chitooligosaccharide (COSM) is derived from the shells of crustaceans such as shrimp and crabs, and the degradation product is obtained by deacetylation of salt. It is formed by glucosamine and n-acetyl glucose linked by a β-1, 4-glycosidic bond. Its degree of polymerization is between 2 and 20, and it is the only cationic basic oligosaccharide [35]. By studying the kinetics of COSM by labeling chitooligosaccharides, it was found that the fluorescence intensity ratio between blood and intestinal contents was 2:1, which proved that most chitooligosaccharides were absorbed into the blood, while the macromolecules of chitooligosaccharides were not absorbed but stayed in the intestine and were directly eliminated from the body over time. Therefore, chitooligosaccharides had an effect on intestinal microorganisms [36]. Because the chitosaccharides of relatively low molecular weight also play a role in release and metabolism in the liver, some of the high-molecular-weight chitosaccharides stay in the intestinal tract and may interact with intestinal microbes and their metabolites to regulate body energy [37]. As prebiotics, chitooligosaccharides can significantly change the abundance of the two main intestinal microbiota, mainly increasing Bacteroidetes [11] and decreasing Firmicutes. Interestingly, when Firmicutes ferments chitooligosaccharides into short-chain fatty acids, some types of short-chain fatty acids are absorbed by the small intestine and provide an energy source for the host [38]. Therefore, it has been speculated that chitooligosaccharides with relatively high polymerization degrees or specific molecular weights can better regulate intestinal microbes. This line of thought was the basis for the current study.

However, recent studies have found that the richness, diversity, and stability of intestinal flora are closely related to host metabolism [39]. When beneficial bacteria are reduced in the microbiome, the balance of the microbiome is disturbed, and the pathogenicity of potentially pathogenic bacteria increases. This is called “microbiome imbalance”. Changes in intestinal flora play an important role in body metabolism. For example, endotoxemia, caused by the passage of the metabolite LPS from intestinal bacteria into the bloodstream through the intestinal barrier, can lead to host metabolic disorders [40]. Therefore, regulating intestinal flora to improve disturbed intestinal flora is a feasible and effective strategy for the prevention and treatment of NAFLD.

According to the literature, gavage of acetic acid can significantly improve liver fat accumulation, enhance mitochondrial function, increase adipose tissue heat production, and reduce body fat in obese mice fed with high-fat diets but has no effect on food intake and skeletal muscle mass [41,42]. This also verifies that when COSM was administered in this paper, it changed the level of acetic acid in short-chain fatty acids. In addition, propionic acid administration in mice improved intestinal gluconeogenesis, controlled glucose homeostasis, and stimulated the release of GLP-1 and PYY in the colon, thereby exerting metabolic benefits [43,44]. Furthermore, oral or intravenous supplementation with SCFAs improves liver fat accumulation and glucose homeostasis by increasing liver AMPK phosphorylation and PPARα gene expression, which are involved in FFA oxidation, glycogen storage, and fat production. Thus, SCFAs play a role in improving nonalcoholic fatty liver disease [44]. Therefore, the therapeutic effect of COSM intervention on NAFLD may also be closely related to SCFAs [45] and other microbial metabolite mediations.

In conclusion, in order to better study the regulatory effect of marine chitosaccharides on intestinal flora, we first induced fatty liver formation in C57 mice with high-fat and high-sugar diets. Then, it was preliminarily confirmed that COSM could improve NAFLD mice to some extent by measuring blood lipids, blood glucose, liver function, inflammatory factors, and antioxidant indexes. In order to explore the causes of the above results, through the determination of intestinal flora and intestinal metabolites, we found that COSM could significantly improve the abundance of beneficial bacteria, reduce the abundance of harmful bacteria, improve the intestinal metabolite total and short chain fatty acid level, and change the structure of intestinal flora and the intestinal bowel barrier, leading to various changes in the regulation of intestinal and flora. Finally, we further analyzed the molecular mechanism of COSM affecting NAFLD and found that the level of serum LPS, which is the connection point of the enterohepatic axis pathway, was significantly reduced under the intervention of COSM. Therefore, to further confirm our research expectations, a WB experiment was used to confirm that COSM could affect the NAFLD process in the LPS/TLR4/NF-κB inflammatory pathway. The results suggested that COSM regulated the LPS/TLR4/NF-κB inflammatory pathway by changing the intestinal flora structure and intestinal barrier and ultimately affected NAFLD progression.

## 4. Materials and Methods

### 4.1. Materials

The COSM used in this study (average MW≤3000 Da, degree of deacetylation ≥ 95%, lot: 180409C) was purchased from Shandong AK Biotech Co., Ltd. (Qingdao, Shandong, China). Metformin was purchased from Sino-American Shanghai Squibb Pharmaceutical Co., Ltd. (Shanghai, China). A normal formula diet (NFD) (20% crude protein, 9.7% moisture, 4.8% crude fiber, 4.3% crude fat, 1.19% calcium, 0.77% phosphorus, and 6.60% crude ash content) was purchased from Guangdong Medical Laboratory Animal Center. A high-fat and high-sucrose diet (HFHSD) (40% and 17% of calories from fat and sucrose, respectively; lot: D1237) was purchased from Research Diet Inc. (New Brunswick, NJ, USA). Information on the antibodies against NF-κB, P-NF-κB, TLR4, ZO-1, occludin, claudin, and GAPDH is provided in Appendix A Table A1.

### 4.2. Animals and Experimental Design

Male C57BL/6 mice were purchased from Slack SJA Laboratory Animal Co., Ltd. (Changsha, China), and raised in the Experimental Animal Center of Guangdong Pharmaceutical University (SYXK (YUE) 2017-0125). The mice were housed in an SPF environment with a temperature of 22–26 °C, 55 ± 5% relative humidity, and a 12 h light/12 h dark cycle. The animal research protocol was approved by the Ethics Committee of the Experimental Animal Center of Guangdong Pharmaceutical University. The animal ethical review number was GDPULACSOF2017378.

Fifty-six 7-week-old C57BL/6 mice (weighing 20–24 g) were fed and used to establish the model and treatment groups in an SPF laboratory. After 1 week of NFD adaptive feeding, all the mice were randomly divided into 2 groups: 8 mice in the control group were given the NFD (based on the feed), and 40 mice in the model group were given the HFHSD. After 8 weeks of feeding, the mice were randomly divided into six groups (each group consisted of 8 mice): blank group (control), NAFLD model group (model), metformin-positive drug group (metformin), COSM high-dose group (COSM-H), COSM medium-dose group (COSM-M), and COSM low-dose group (COSM-L).

Based on formula conversion and preliminary research conducted by our research group, the dosages of COSM used in this experiment were as follows: COSM-H, 1700 mg/kg/d; COSM-M, 850 mg/kg/d; and COSM-L, 425 mg/kg/d. According to a previous study [46], the dosage administered to the metformin-positive drug control group in this experiment was determined to equal 50 mg/kg/d. According to the body weight of the mice, the doses administered to the COSM-H, COSM-M, and COSM-L groups and the amount of dimethyl dicarbonate were adjusted to 10 g/0.1 mL. The mice in the control group were also given the same volume of ultrapure water at a dose of 10 g/0.1 mL.

During the animal experiment, the mice were weighed every week. The weight of the remaining feed and the weight of the feed given to each group of mice were measured every day to estimate their daily food intake. The physical and mental conditions of the mice were observed at each timepoint when data were recorded. After the 12-week intervention, the mice were anesthetized by isoflurane inhalation, and blood samples were collected by orbital blood sampling. Blood samples; samples of the liver, ileum, colon, and cecal contents; and feces were collected and stored at −80 °C.

### 4.3. Serum and Liver Index Analyses

For serum index determination, blood samples were maintained at room temperature for 30 min and centrifuged at 4 °C and 3000 rpm for 15 min. The upper serum was collected, and the serum TC, TG, HDL-C, LDL-C, and FFA levels were determined using a kit (Nanjing Jiancheng Bioengineering Institute, Nanjing, China) according to the instructions. The AST, ALT, CAT, and T-AOC levels and the serum levels of IL-6, IL-10, TNF-α, insulin, and LPS were determined using enzyme-linked immunosorbent assay (ELISA) kits (MEIMIAN, Yancheng, China).

For liver index measurement, an equal amount of liver tissue was accurately weighed, an appropriate amount of absolute ethanol was added, and the samples were homogenized with steel balls in a homogenizer precooled at −20 °C. The samples were subsequently centrifuged at 4 °C and 2500 rpm for 10 min, the supernatant was collected, and Box reagents (Nanjing Jiancheng Bioengineering Institute, Nanjing, China) were used to determine the TG and TC contents.

### 4.4. Histopathological Analysis

For hematoxylin and eosin (H&E) staining, the liver, ileum, and colon tissues were fixed in 4% paraformaldehyde (PFA), dehydrated by gradient ethanol, permeabilized with xylene, soaked in high-melting-point paraffin, embedded in paraffin, sliced to a thickness of 4 μm, and stained with H&E (Leagene Biotechnology, Beijing, China). The fatty degeneration of liver cells and the damage to intestinal epithelial cells were observed.

For oil red O staining, liver tissue was fixed overnight in 4% PFA, dehydrated through a sucrose solution gradient, embedded in optimal cutting temperature (OCT) compound (Sakura), cut into 7 μm-thick frozen sections, and stained with oil red O (Sigma, St. Louis, MO, USA) for the observation of lipid droplets in liver cells.

### 4.5. Fluorescence Quantitative PCR (RT-PCR)

The primers of TNF-α, IL-6, and IL-10 in mice were determined by reviewing relevant literature and comparing the data obtained herein. All primers were synthesized by Shanghai Shenggong Biological Co., Ltd. Primer sequences can be seen in Appendix A Table A3. (1) Total RNA was extracted and 0.1 g tissue was weighed and placed in a 1.5 mL enzyme-free centrifuge tube. Before adding tissue into the centrifuge tube, two steel balls were added and placed in the refrigerator for precooling. Then, 1 mL RNAiso Plus (total RNA extraction reagent) was added to the centrifuge tube and homogenized in the homogenizer for 10 min. Note that the whole process was carried out on ice. (2) CDNA was synthesized by removing genomic DNA and reverse transcription. (3) To carry out real-time fluorescence quantitative polymerase chain reaction analysis, reverse transcripted cDNA and synthesized primers of corresponding genes were taken and added to a Roche 96-well PCR plate with the following reagents (TaKaRa TB Green Premix Ex Taq^TM^ II kit including Upstream primer, downstream primer, TB Green Premix Ex Taq^TM^ II, cDNA and DEPC water) in the corresponding quantities. After the amplification reaction, the data were recorded, the relative quantitative expression of genes between groups was analyzed by the 2^−ΔΔCt^ method, and the relative expression levels were calculated.

### 4.6. Fecal Microbiota 16S rRNA Analysis

Next, 16S rRNA (V3–V4 region) sequencing was performed to analyze the composition of the intestinal microbiota in feces. Metware Biotechnology (Wuhan, China) was used to perform the sequencing and analysis in this study. The specific experimental method was previously described by Li et al. [47]. See Appendix A Table A2 for primer information for the 16S rRNA sequence.

The CTAB/SDS method was used for the extraction of total genomic DNA from feces (Sambrook and Russell, 2001), and the resulting DNA was used as the template for PCR amplification of the V3–V4 variable region using universal primers (Table A2). Equal amounts of purified amplicons were combined for subsequent sequencing with the Ion S5™ XL platform (Thermo Fisher, Waltham, MA, USA).

Quality filtering of the raw tags was performed under specific filtering conditions to obtain high-quality clean tags according to QIIME (Version 1.9.1). Sequences with ≥97% similarity were assigned to the same operational taxonomic units (OTUs). The abundance information of the OTUs was normalized using a standard sequence number corresponding to the sample with the lowest number of sequences.

Various analyses, including alpha and beta diversity calculations, were performed using the QIIME software package, and the alpha diversity analysis included calculations of the ACE, Chao1, Shannon, and Simpson diversity indexes. The weighted UniFrac distance metric was used for principal coordinates analysis (PCoA) to visualize the separation of the samples. Beta diversity analysis was performed to evaluate the differences in the species complexity of the samples.

### 4.7. Short-Chain Fatty Acid (SCFA) Profile Analysis

Twenty milligrams of the cecal content samples was weighed and placed in a 2 mL Eppendorf (EP) tube, and 1 mL of phosphoric acid (0.5% *v/v*) solution was added to the EP tube. The tubes were then vortexed for 10 min and ultrasonicated for 5 min. Subsequently, 0.1 mL of supernatant was added to a 1.5 mL centrifuge tube, and 0.5 mL of methyl tert-butyl ether (MTBE) (containing internal standard) solution was added. The samples were vortexed for 3 min, ultrasonicated for 5 min, and centrifuged for 10 min at 12,000 r/min and 4 °C. After centrifugation, 0.2 mL of supernatant was absorbed into the sampling bottle for GC–MS/MS analysis.

### 4.8. Western Blotting

Western blot analysis was performed as previously described [48] to analyze the protein expression levels of ZO-1, occludin, and claudin in the colon and of Toll-like receptor 4 (TLR4), nuclear factor kappa B (NF-κB), and phospho-nuclear factor kappa B p65 (P-NF-κB p65) in the liver. The primary and secondary antibodies are summarized in Table A1. The Gel DocXR+ gel imaging system (Bio–Rad Laboratories, Hercules, CA, USA) was used to image the polyvinylidene fluoride (PVDF) membrane, and the densities of the bands were analyzed with the Bio–Rad Image Lab (Ver. 6.0) software. The relative expression levels of the proteins were normalized to those of GAPDH.

### 4.9. Statistical Analysis

The data were analyzed and graphed using the software SPSS 20.0 and GraphPad Prism 8.3 (GraphPad Software, San Diego, CA, USA) by one-way ANOVA for quantitative multigroup comparisons, as shown in the figure legends. The error bars indicate the standard deviations (SDs) obtained from all statistical analyses. The main results are displayed as bar graphs, and *p* < 0.05 was considered to indicate statistical significance.

## 5. Conclusions

An NAFLD mouse model was induced by HFHSD, and the effects of different doses of COSM gavage were compared with that of metformin. It was found that marine chitooligosaccharide (COSM) could improve IR and liver steatosis, reduce the level of lipid index, decrease the accumulation of lipid in the liver, enhance the antioxidant capacity of the liver, and alleviate liver damage and inflammation without affecting appetite in mice. These results suggest that COSM has great potential as a drug candidate for NAFLD. Subsequently, we found that COSM affected the intestinal flora structure, reduced the F/B ratio, reversed an increase in bacterial diversity, increased the abundance of some beneficial bacteria, reduced the abundance of some pathogenic bacteria, improved the abnormal intestinal barrier structure, and reversed the changes of the flora caused by HFHSD in NAFLD mice. Moreover, the SCFA profile of intestinal metabolites was found to play an important role in the treatment of NAFLD and have a certain dose dependence. Furthermore, it was found that COSM reduced LPS translocation due to intestinal flora, improved the expression of intestinal barrier function-related proteins, and ultimately reduced the expression level of the liver LPS/TLR4/NF-κB signaling pathway, thereby improving liver disease in NAFLD mice. This information provides not only a theoretical basis and reference for the development of new NAFLD drugs but a basis for in-depth understanding of the molecular mechanism of COSM. It also provides a reference basis and broad development prospects for the research of marine pharmaceutical products.

## Figures and Tables

**Figure 1 marinedrugs-20-00383-f001:**
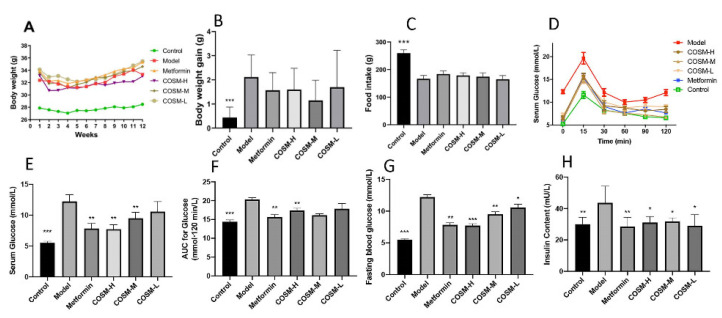
Changes in body weight, food intake, and serum glucose and lipid levels. (**A**) Body weight changes of HFHSD mice during 12 weeks of oral COSM administration. (**B**) Changes in body weight gain of HFHSD mice after 12 weeks of oral administration. (**C**) Dietary intake of HFHSD mice during 12 weeks of oral administration. (**D**–**F**) HFHSD mice were administered COSM orally for 12 weeks after oral glucose tolerance tests. (**G**) Glucose levels in HFHSD mice after 12 weeks of oral administration of COSM. (**H**) Insulin levels in HFHSD mice after 12 weeks of oral administration of COSM. COSM could improve IR and hyperglycemia without affecting food intake in mice. The data are presented as means ± SDs (*n* = 8). Compared with the model group, * *p* < 0.05, ** *p* < 0.01, *** *p* < 0.001.

**Figure 2 marinedrugs-20-00383-f002:**
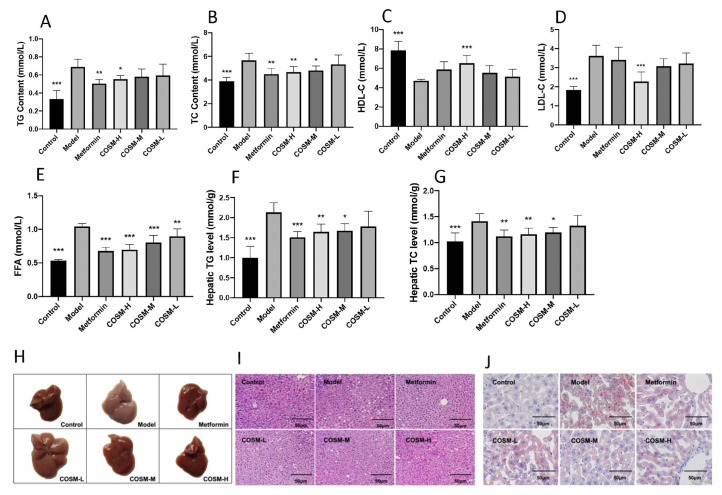
COSM could improve the serum and liver lipid contents of HFHSD-fed mice. The following indicators were measured after 12 weeks of oral administration of COSM in HFHSD mice: (**A**) serum TG level; (**B**) serum TC level; (**C**) serum HDL-C level; (**D**) serum LDL-C level; (**E**) serum FFA level; (**F**) liver TG level; (**G**) liver TC level; (**H**) fresh liver samples; (**I**) H&E staining; (**J**) oil red O staining. These results suggest that COSM can improve the lipid metabolism abnormalities caused by an HFHSD to a certain extent and reverse lipid accumulation in the livers of mice with NAFLD to a certain extent. The data are presented as means ± SDs (*n* = 8). Compared with the model group, * *p* < 0.05, ** *p* < 0.01, *** *p* < 0.001.

**Figure 3 marinedrugs-20-00383-f003:**
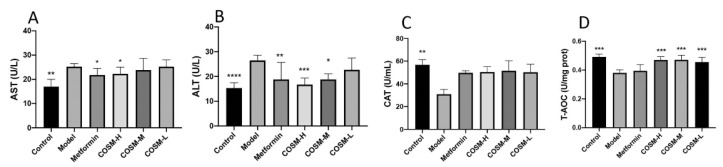
COSM could improve the liver function and antioxidant capacity of HFHSD-fed mice. The following indicators were measured after 12 weeks of oral administration of COSM in HFHSD mice: (**A**) serum AST level; (**B**) serum ALT level; (**C**) serum CAT level; (**D**) serum T-AOC level. These results suggest that COSM can improve the abnormal liver function induced by HFHSD and enhance the antioxidant function of mice with NAFLD to a certain extent. The data are presented as means ± SDs (*n* = 8). Compared with the model group, * *p* < 0.05, ** *p* < 0.01, *** *p* < 0.001, **** *p* < 0.0001.

**Figure 4 marinedrugs-20-00383-f004:**
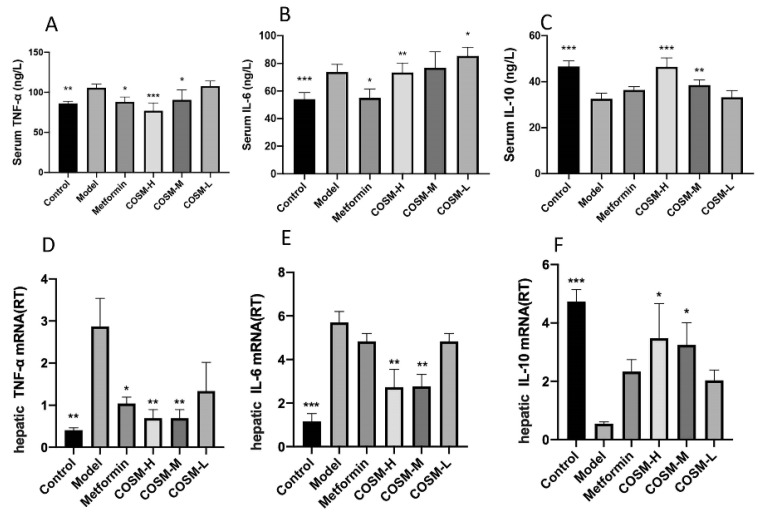
COSM ameliorates serum inflammation in HFHSD-fed mice. The following indicators were measured after 12 weeks of oral administration of COSM in HFHSD mice: (**A**) serum TNF-α level; (**B**) serum IL-6 level (**C**) serum IL-10 level; (**D**) TNF-α gene expression levels in the liver; (**E**) Il-6 gene expression levels in the liver; (**F**) Il-10 gene expression levels in liver. These results suggest that COSM can improve the serum inflammatory response of HFHSD-fed mice. The data are presented as means ± SDs (*n* = 8). Compared with the model group, * *p* < 0.05, ** *p* < 0.01, *** *p* < 0.001.

**Figure 5 marinedrugs-20-00383-f005:**
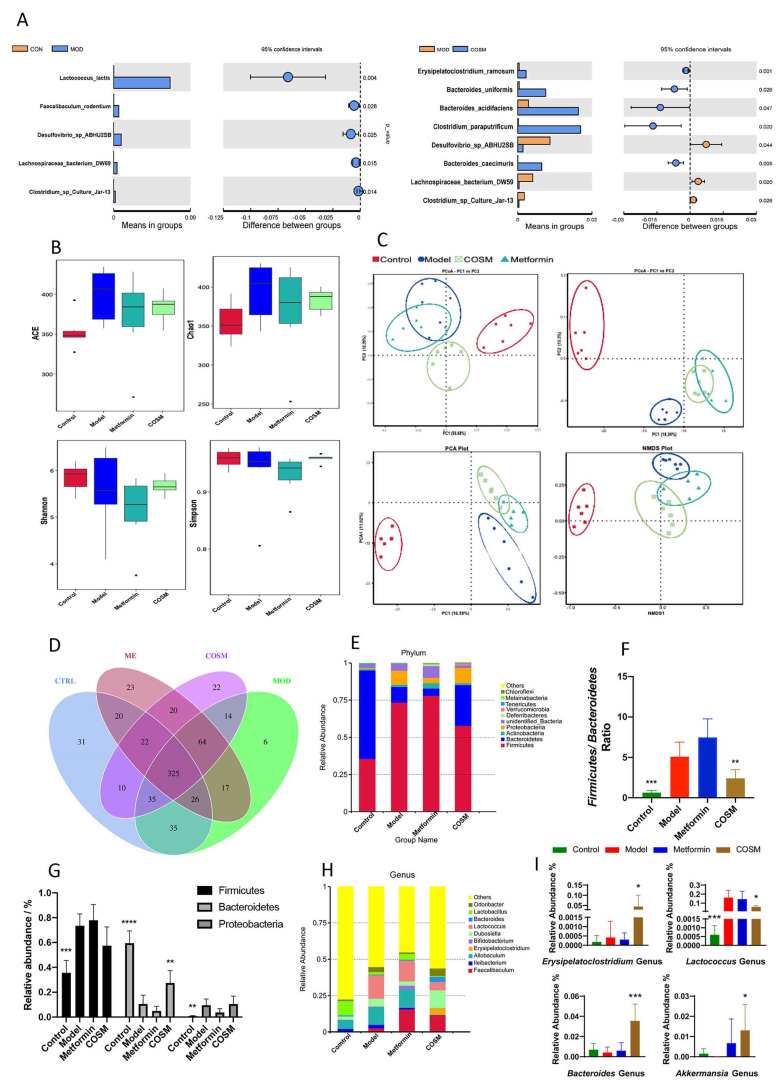
COSM could improve the intestinal microflora structure of HFHSD-fed mice, inhibit the growth of harmful intestinal bacteria in HFHSD-fed mice, and increase the abundance of dominant bacteria in HFHSD-fed mice. The following indicators were measured after 12 weeks of oral administration of COSM in HFHSD mice: (**A**) T-test analysis at the species level; (**B**) alpha diversity analysis (ACE, Chao1, Shannon, and Simpson indexes); (**C**) beta diversity analysis (PCoA, principal component analysis (PCA), and nonmetric multidimensional calibration (NMDS)); (**D**) fecal sample sequencing to analyze the OTU values of each group; (**E**) gate level analysis; (**F**) ratio analysis of Bacteroidetes and Firmicutes; (**G**) abundance of Bacteroidetes, Firmicutes, and Proteobacteria in the feces of mice in each group; (**H**) genus-level analysis; (**I**) quantitative analysis of dominant genera. The data are presented as means ± SDs (*n* = 6). Compared with the model group, * *p* < 0.05, ** *p* < 0.01, *** *p* < 0.001, **** *p* < 0.0001.

**Figure 6 marinedrugs-20-00383-f006:**
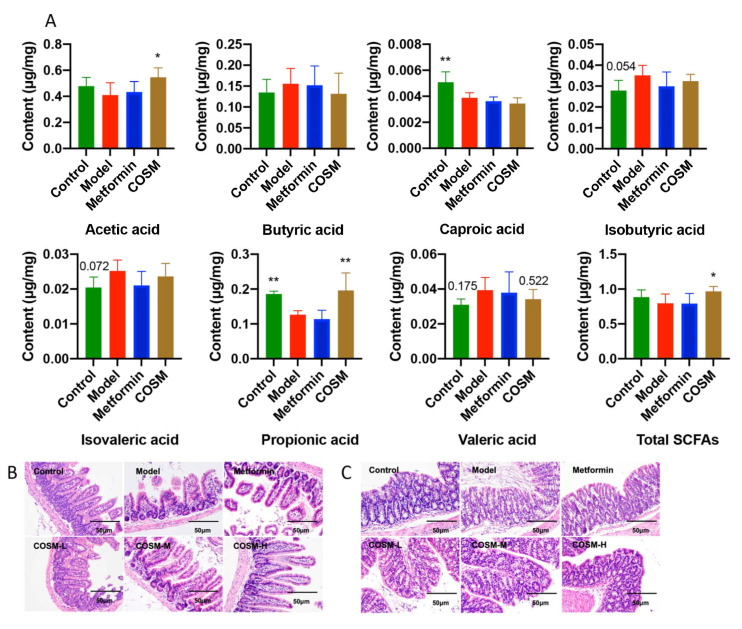
COSM increased the levels of acetic acid, propionic acid, and total SCFAs in the cecal contents of HFHSD-fed mice and improved the villus structure of the ileum and the integrity of colonic goblet cells and the mucosal layer of HFHSD-fed mice. The following indicators were measured after 12 weeks of oral administration of COSM in HFHSD mice. (**A**) SCFA analysis; (**B**) H&E staining of the ileum; (**C**) H&E staining of the colon. The data are presented as means ± SDs (*n* = 6). Compared with the model group, * *p* < 0.05, ** *p* < 0.01.

**Figure 7 marinedrugs-20-00383-f007:**
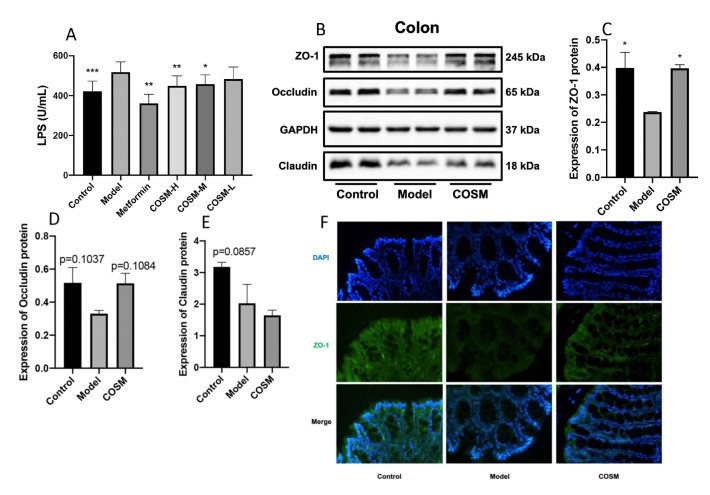
COSM improved the intestinal wall barrier integrity and endotoxemia in HFHSD-fed mice. The following indicators were measured after 12 weeks of oral administration of COSM in HFHSD mice: (**A**) serum LPS level; (**B**) WB images of ZO-1, occludin, and claudin protein expression; (**C**) colon ZO-1 protein expression level; (**D**) occludin protein expression level in the colon; (**E**) claudin protein expression in the intestinal mucosa; (**F**) colon ZO-1 immunofluorescence staining. The data are presented as means ± SDs (*n* = 6). Compared with the model group, * *p* < 0.05, ** *p* < 0.01, *** *p* < 0.001.

**Figure 8 marinedrugs-20-00383-f008:**
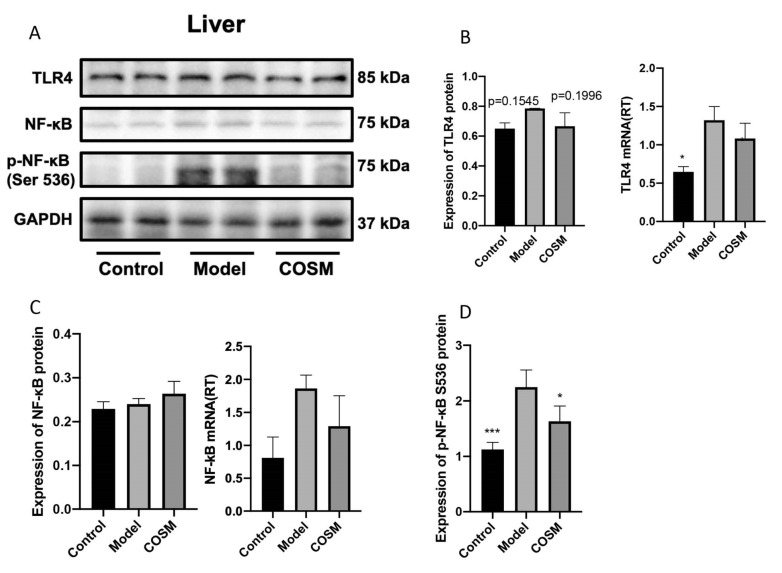
COSM improved the LPS/TLR4/NF-κB signaling pathway in the livers of HFHSD-fed mice. The following indicators were measured after 12 weeks of oral administration of COSM in HFHSD mice: (**A**) WB images of TLR4, NF-κB, and P-NF-κB protein expression; (**B**) TLR4 protein expression level and gene expression level in the liver; (**C**) expression level of NF-κB protein and gene in the liver; (**D**) protein expression of P-NF-κB in the liver. The data are presented as means ± SDs (*n* = 6). Compared with the model group, * *p* < 0.05, *** *p* < 0.001.

## Data Availability

The data presented in this study are available within the article.

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
