# Peer review of "Marine Chitooligosaccharide Alters Intestinal Flora Structure and Regulates Hepatic Inflammatory Response to Influence Nonalcoholic Fatty Liver Disease"

_marinedrugs, 2022, doi:10.3390/md20060383_

Round 1

Reviewer 1 Report

It is an interesting study wherein the authors have investigated the role of chitooligosaccharide in reducing hepatic steatosis induced by HFHSD-fed mice. They showed that COSM treatment can alter intestinal flora structure and activate the inflammatory signaling pathway. The experimental protocol is well designed and the discussion is well supported by the data. 

Following are comments to further improve the quality of the article;

  1. A structured abstract is not required. The abstract should be in a single paragraph. Plz refer to the latest issue of the journal and instructions to the authors.
  2. Aim: Line 20: Rephrase it to convey the meaning.
  3. Line 23- "COSM solution was intragastric for 12 weeks" is an incomplete sentence. 
  4. line 34: "Downregulation ............pathway" is an incomplete sentence.
  5. line 49: Separate the two words in abovementioned
  6. lines 52-54: "in recent years....... chitosan" Rephrase it, as it does not convey the intended meaning.
  7. line 78: Delete COS and write COSM without brackets.
  8. Sec 2.3: line 163-164: Indicate % increase in ALT and AST levels in Sham with respect to control.
  9. Sec 2.4: line 186: Write HFHSD in place of high fat and high sugar diet. 
  10. lines 425 and 430: you have written 20 and 33. Are these references? write them properly.

Author Response

Dear editors and reviewers,

On behalf of my co-authors, we thank you very much for giving us an opportunity to revise our manuscript, we appreciate editor and reviewers very much for their positive and constructive comments and suggestions on our manuscript entitled “Marine chitooligosaccharide alter intestinal flora structure and regulate hepatic inflammatory response to influence non-alcoholic fatty liver disease” (Manuscript ID: marinedrugs-1735106). We have studied reviewer’s comments carefully and have made revision which marked in the “Track Changes” function in the paper. We would like to submit the revised manuscript. The responses to the reviewers’ comments are provided below.

Responses to Reviewer 1 comments:

Thank you very much for your constructive comments.

Point 1: A structured abstract is not required. The abstract should be in a single paragraph. Plz refer to the latest issue of the journal and instructions to the authors.

Response 1: Thank you for your useful comments and suggestions. We are very sorry for our errors. We have updated the previous summary to make it unstructured.

Point 2: Aim: Line 20: Rephrase it to convey the meaning.

Response 2: Thank you very much. We are very sorry for our errors. We rechecked the purpose, deleted it, and clarified the purpose of this article at the end of the article.

Point 3: Line 23- "COSM solution was intragastric for 12 weeks" is an incomplete sentence.

Response 3: Thank you for your useful comments and suggestions. The wrong sentence has been changed to a more complete meaning of " In this study, C57BL/6 mice were given HFHSD diet for 8 weeks to induce hepatic steatosis, and then given COSM solution orally for 12 weeks."

Point 4: line 34: "Downregulation ............pathway" is an incomplete sentence.

Response 4: Thank you for your useful comments and suggestions. The wrong sentence has been changed to a more complete meaning of " In addition, COSM not only improved LPS levels and barrier dysfunction in ileum and colon, but also up-regulated protein levels of ZO-1, Occludin and Claudin in colon, and down-regulated liver LPS/TLR4/NF-κB inflammatory pathway."

Point 5: line 49: Separate the two words in abovementioned

Response 5: Thank you for your useful comments and suggestions. The wrong word "abovementioned" on line 49 has been changed to "above mentioned."

Point 6: lines 52-54: "in recent years....... chitosan" Rephrase it, as it does not convey the intended meaning.

Response 6: Thank you for your useful comments and suggestions. "In recent years....... Chitosan "this sentence has been rewritten as" Marine resources ……have been studied extensively."

Point 7: line 78: Delete COS and write COSM without brackets.

Response 7: Thank you for your useful comments and suggestions. The COS and () symbols have been removed from this article.

Point 8: Sec 2.3: line 163-164: Indicate % increase in ALT and AST levels in Sham with respect to control.

Response 8: Thank you for your useful comments and suggestions. " Indicate % increase in ALT and AST levels in Sham with respect to control."this sentence has been rewritten as “The serum AST and ALT levels in model group were 48.70% and 72.67% higher than those in control group, respectively, suggesting that long-term HFHSD feeding may damage the liver function of model group to a certain extent.”

Point 9: Sec 2.4: line 186: Write HFHSD in place of high fat and high sugar diet.

Response 9: Thank you for your useful comments and suggestions. Line 186 "high - fat and high - sugar diet" has been changed to "HFHSD."

Point 10: lines 425 and 430: you have written 20 and 33. Are these references? write them properly.

Response 10: Thank you for your useful comments and suggestions. Article 20 and 33 are references and have been updated as references [39] and [40].

Reviewer 2 Report

Feng et al. usefulness of marine chitooligosaccharide as therapeutics against NAFLD by altering intestinal flora structure. The paper is well-written and organized. However, some findings are difficult to understand due to a lack of explanation. Here are the comments.

The first figure Fig. 1A is hard to read. Was there bodyweight difference in just one week's high-fat diet? Please clarify this in the legend. Regarding Figure 1B, please explain the period of body weight gain. The same issue is in Figure 1C. A tolerance test should be mentioned in the method section. It is hard to understand the time when the authors measure the glucose levels.

Figure 2. The data were analyzed and graphed using SPSS 20.0 and GraphPad Prism 7.0 software. This sentence should be moved to the method section. The paired comparison is unclear. The authors should use a multiple comparison model or explain which pair they compared.

Figure 3. Again, it is unclear the time they analyzed the sample.

Since there is no significance in the CAT level, the authors cannot state that COSM can enhance the antioxidant function.

Figure 4. Please include the number of mice in each graph or use a dotted plot instead.

The review would think the authors could evaluate the cytokine levels in the liver by Western blot or ELISA.

Figure 5. Please include control mice flora in the analysis (A).

Figure 6. Please discuss the relevance of acetic acid in the discussion section.

Figure 7. The quality of immunofluorescence imaging is poor please replace them with better pictures or with higher magnification.

Figure 8. LPS/TLR4/NF-κB pathway analysis should be performed with mRNA analysis.

Author Response

Dear editors and reviewers,

On behalf of my co-authors, we thank you very much for giving us an opportunity to revise our manuscript, we appreciate editor and reviewers very much for their positive and constructive comments and suggestions on our manuscript entitled “Marine chitooligosaccharide alter intestinal flora structure and regulate hepatic inflammatory response to influence non-alcoholic fatty liver disease” (Manuscript ID: marinedrugs-1735106). We have studied reviewer’s comments carefully and have made revision which marked in the “Track Changes” function in the paper. We would like to submit the revised manuscript. The responses to the reviewers’ comments are provided below.

Responses to Reviewer 2 comments:

Thank you very much for your constructive comments.

Point 1: The first figure Fig. 1A is hard to read. Was there bodyweight difference in just one week's high-fat diet? Please clarify this in the legend. Regarding Figure 1B, please explain the period of body weight gain. The same issue is in Figure 1C. A tolerance test should be mentioned in the method section. It is hard to understand the time when the authors measure the glucose levels.

Response 1: Thank you very much. We are very sorry for our errors. The questions raised by the experts are explained in the result 2.1 of the text and in Figure1.

Point 2: Figure 2. The data were analyzed and graphed using SPSS 20.0 and GraphPad Prism 8.3 software. This sentence should be moved to the method section. The paired comparison is unclear. The authors should use a multiple comparison model or explain which pair they compared.

Response 2: Thank you very much. We are very sorry for our errors. The sentence "analyze and plot data using SPSS 20.0 and GraphPad Prism 8.3 software" has been moved to the methods section. In addition, the comparison objects have been pointed out in the graph notes, and all the results presented are compared with the Model group.

Point 3: Figure 3. Again, it is unclear the time they analyzed the sample. Since there is no significance in the CAT level, the authors cannot state that COSM can enhance the antioxidant function.

Response 3: Thank you for your useful comments and suggestions. It has been pointed out in the graph notes of the paper that "it is not clear when the results determine the analysis samples". Regarding "since CAT levels are not significant, the authors cannot say that COSM enhances antioxidant function." In my opinion, although there was no significant difference in CAT level after COSM oral administration for 12 weeks, there was significant difference in T-AOC level between coSM-H group and COSM-M group. Therefore, I believe that COSM can enhance antioxidant function to some extent in combination with CAT and T-AOC levels. This is one of the reasons why this paper chose inflammatory signaling pathway rather than antioxidant capacity for further research.

Point 4: Figure 4. Please include the number of mice in each graph or use a dotted plot instead. The review would think the authors could evaluate the cytokine levels in the liver by Western blot or ELISA.

Response 4: Thank you very much. We are very sorry for our errors. The number of mice in each diagram is 6, that is, n = 6. Western blot and ELISA were not supplemented in this paper, but liver QT-PCR was supplemented, because in the author's opinion, both QT-PCR and Western blot and ELISA are semi-quantitative methods to detect indicators. Therefore, QT-PCR experiment was added in this paper.

Point 5: Figure 5. Please include control mice flora in the analysis (A).

Response 5: Thank you very much. We are very sorry for our errors. The analysis of the gut microbiota of the CON group in Figure 5A has been supplemented.

Point 6: Figure 6. Please discuss the relevance of acetic acid in the discussion section.

Response 6: Thank you very much. We are very sorry for our errors. The correlation of acetic acid has been discussed in the discussion section. The importance of acetic acid is also elaborated in 2.5.3.

Point 7: Figure 7. The quality of immunofluorescence imaging is poor please replace them with better pictures or with higher magnification.

Response 7: Thank you very much. We are very sorry for our errors. Regarding the question "immunofluorescence imaging quality is poor, please replace them with better pictures or higher magnification", we have updated the pictures with better pixels.

Point 8: Figure 8. LPS/TLR4/NF-κB pathway analysis should be performed with mRNA analysis.

Response 8: Thank you very much. We are very sorry for our errors. The question "LPS/TLR4/NF-κB pathway analysis should be performed with mRNA analysis" has been added in Figure 8.

Reviewer 3 Report

The study is part of a very crowded and very interesting trend and I recommend publication.

There are some details to be corrected:

1. reference 1: it would be better to cite an epidemiological study.

2. reference 4 is inappropriate

3. There are more appropriate reference instead of 14

4. line 95: I didn't find section 3.5.2

5. line 98: "F/B" is defined later (should be done here)

6. line 114: TC serum is not defined

7. line 168 does "increase" means "improve"?

8. line 271: "not significant" should be added before decreased abundance of Firmicutes

9. line 355: the claudin was omitted.

10: line 385 IR has not been defined

11. figure 5 is wrong (twice "C")

12: line 459 "you"?

Author Response

Response to Reviewer 3 Comments

Dear editors and reviewers,

On behalf of my co-authors, we thank you very much for giving us an opportunity to revise our manuscript, we appreciate editor and reviewers very much for their positive and constructive comments and suggestions on our manuscript entitled “Marine chitooligosaccharide alter intestinal flora structure and regulate hepatic inflammatory response to influence nonalcoholic fatty liver disease” (Manuscript ID: marinedrugs-1735106). We have studied reviewer’s comments carefully and have made revision which marked in the “Track Changes” function in the paper. We would like to submit the revised manuscript. The responses to the reviewers’ comments are provided below.

Responses to Reviewer 3 comments:

Thank you very much for your constructive comments.

Point 1: reference 1: it would be better to cite an epidemiological study.

Response 1: Thank you very much. We are very sorry for our errors. The issue of "Reference 1: It is best to cite epidemiological studies" has been addressed in an updated version of reference [1].

Point 2: reference 4 is inappropriate.

Response 2: Thank you very much. We are very sorry for our errors. Reference 4 has been corrected.

Point 3: There are more appropriate reference instead of 14.

Response 3: Thank you for your useful comments and suggestions. Reference 14 has been corrected.

Point 4: line 95: I didn't find section 3.5.2.

Response 4: Thank you very much. We are very sorry for our errors. 3.5.2 has been corrected to 2.5.2.

Point 5: line 98: "F/B" is defined later (should be done here).

Response 5: Thank you very much. We are very sorry for our errors. This issue has been clarified in the updated manuscript 2.1.

Point 6: line 114: TC serum is not defined.

Response 6: Thank you very much. We are very sorry for our errors. This issue has been updated in the latest manuscript 2.2.

Point 7: line 168 does "increase" means "improve"?

Response 7: Thank you very much. We are very sorry for our errors. The word "increase" in line 168 means "improve ", but the word "reduced" is used to prevent ambiguity or misunderstanding. Specific lines are based on the latest version of the manuscript.

Point 8: line 271: "not significant" should be added before decreased abundance of Firmicutes.

Response 8: Thank you very much. We are very sorry for our errors. The question you asked has been changed in the latest version, on line 488.

Point 9: line 355: the claudin was omitted.

Response 9: Thank you very much. We are very sorry for our errors. The issue you raised has been changed in the latest version 2.6.

Point 10: line 385 IR has not been defined.

Response 10: Thank you very much. We are very sorry for our errors. The IR issue you raised has been defined above in the latest version.

Point 11: figure 5 is wrong (twice "C").

Response 11: Thank you very much. We are very sorry for our errors. The problem you raised has been corrected in the previous statement in the latest edition.

Point 12: line 459 "you"?

Response 12: Thank you very much. We are very sorry for our errors. This "you" is extra and has been deleted. This "you" refers to "COSM".

Round 2

Reviewer 2 Report

The authors have modified the manuscript as requested. I have no further comments.

Author Response

Dear reviewer, I would like to thank you very much for your review of this manuscript, which has given me an opportunity to improve and make progress. I would like to express my high gratitude for your review opinions once again.